# Carbonyl Composition and Electrophilicity in Vaping Emissions of Flavored and Unflavored E-Liquids

**DOI:** 10.3390/toxics9120345

**Published:** 2021-12-09

**Authors:** Jin Y. Chen, Alexa Canchola, Ying-Hsuan Lin

**Affiliations:** 1Environmental Toxicology Graduate Program, University of California, Riverside, CA 92521, USA; jychen03@louisville.edu (J.Y.C.); acanc007@ucr.edu (A.C.); 2Department of Environmental Sciences, University of California, Riverside, CA 92521, USA

**Keywords:** carbonyl emissions, alcohol-containing flavoring chemicals, propylene glycol, vegetable glycerin, *trans*-2-hexenol, benzyl alcohol, l-(-)-menthol, linalool, global electrophilicity, condensed Fukui parameters

## Abstract

It has been demonstrated that propylene glycol (PG), vegetable glycerin (VG), and flavoring chemicals can thermally degrade to form carbonyls during vaping, but less is known about carbonyl emissions produced by transformation of flavoring chemicals and the interactive effects among e-liquid constituents. This study characterized carbonyl composition and levels in vaping emissions of PG-VG (e-liquid base solvents) and four e-liquid formulations flavored with *trans*-2-hexenol, benzyl alcohol, l-(-)-menthol, or linalool. Utilizing gas chromatography (GC)- and liquid chromatography (LC)-mass spectrometry (MS) methods, 14 carbonyls were identified and quantified. PG-VG emitted highest levels of formaldehyde, acetaldehyde, and acrolein. However, flavored e-liquids contributed to the production of a wider variety of carbonyls, with some carbonyls directly corresponding to the oxidation of alcohol moieties in flavoring compounds (e.g., *trans*-2-hexenol and benzyl alcohol transformed into *trans*-2-hexenal and benzaldehyde, respectively). Detections of formaldehyde-GSH and *trans*-2-hexenal-GSH adducts signify interactions of carbonyls with biological nucleophiles. The global reactivity descriptors (*I*, *A*, *μ*, *η*, and *ω*) and condensed Fukui parameters ($f_{k}^{0}$, $f_{k}^{-}$, $f_{k}^{+}$, and dual-descriptor) were computed to elucidate site reactivities of selected simple and α,β-unsaturated carbonyls found in vaping emissions. Overall, this study highlights carbonyl emissions and reactivities and their potential health risk effects associated with vaping.

## 1. Introduction

E-cigarettes are new and emerging tobacco products that use heat-not-burn electronic devices to deliver aerosolized e-liquids containing nicotine, base solvents (e.g., propylene glycol (PG) and vegetable glycerin (VG)), flavoring chemicals, and/or other ingredients into the users via inhalation. Many smokers have used e-cigarettes to help with smoking cessation [1], but the use of e-cigarettes among non-smoking teenagers and young adults is prevalent in the United States [2]. One factor contributing to the widespread popularity of e-cigarettes among non-smoking teenagers and adults alike is the use of flavored e-liquids, which come in a variety of flavors, including fruit, menthol, mint, and tobacco [3,4]. While e-cigarettes are frequently marketed as a safer alternative to conventional tobacco products, the e-cigarette vaping-associated health outcomes have not been fully investigated. Recent research has found a link between e-cigarette vaping and illnesses, such as lung injury [5] and oral health problems [6], indicating that vaping is not without risk.

Flavoring chemicals used in e-liquids not only appeal to users, but they may contribute to adverse health effects associated with e-cigarette vaping. For example, cinnamaldehyde is regarded as safe for ingestion, but cinnamaldehyde found in e-liquids and vaping aerosols has been found to cause cytotoxicity [7], to suppress immune functions of respiratory tract cells [8], and to disrupt mitochondrial function of human bronchial epithelial cells [9]. Degradation of e-liquid constituents, such as PG, VG, and flavoring chemicals, can produce reactive compounds in vaping emissions. Studies have demonstrated that thermal decomposition of PG, VG, and flavoring chemicals can release reactive carbonyl species, including formaldehyde, acetaldehyde, acrolein, and propionaldehyde [10,11,12,13]. Among commonly reported carbonyls, formaldehyde is a known human carcinogen that can cause a wide range of health implications, such as irritations and pharyngeal cancer, through inhalation exposure [14].

Carbonyls are reactive electrophiles that can form adducts with biological nucleophiles, such as glutathione (GSH) [15], protein residues [16,17], and DNA [18,19]. For example, as an α,β-unsaturated carbonyl, acrolein can covalently modify proteins by forming adducts with the cysteine residue via Michael addition and Schiff base formation, as well as other nucleophilic side chains (e.g., the histidine and lysine residues) and free amino terminus of proteins [20]. Reactions of carbonyls toward cellular nucleophiles can result in a wide range of adverse health effects, including oxidative DNA breakage [21], formation of protein adducts [22], decrease or loss of protein functions [23], and eventually pathogenesis. Exposure to exogenous carbonyls from cigarette smoke has been linked to DNA adduct formation and carcinogenesis [24]. Wang et al. [25] detected an elevated level of formaldehyde-DNA adduct (N6-hydroxymethyldeoxyadenosine) in human leukocytes of smokers compared to nonsmokers, indicating that formaldehyde is a contributing factor of smoking-induced cancer.

To model reactivity and toxicity of various carbonyls associated with vaping, computational chemistry methods, such as density functional theory and condensed Fukui functions, have been used to study global electrophilicity [26] and chemical reactivity at specific sites within a molecule [27,28]. Wondrousch et al. [29] employed global electrophilicity index (ω) and local site-specific parameters (energy-weighted local electrophilicity, ω^E^) and local charge-limited local electrophilicity (ω^q^) to predict reaction rates of 31 α,β-unsaturated carbonyls toward GSH. Global and local electrophilicity parameters correlated well with experimental values with *r*^2^ values around 0.88 and 0.89, respectively, which demonstrate sufficient applicability of computational approaches to predict carbonyls’ chemical reactivity [29]. Schwöbel et al. [30] developed a model that uses ω^q^ at the β-carbon of Michael-type acceptors to predict the kinetic rate constants of 66 compounds (aldehydes and ketones) toward GSH (k_GSH_), and predicted k_GSH_ values correlated well with experimental rate constants with *r*^2^ of 0.91. Chen and Jiang et al. [28] correlated ω values of 10 carbonyls with experimental consumption rates of dithiothreitol (k_DTT_), a commonly used assay to estimate the oxidative potential of atmospheric aerosols [31]. The findings revealed a strong relationship between ω and k_DTT_ values with approximate *r*^2^ values of 0.84 and 0.99 for simple and α,β-unsaturated carbonyls, respectively [28].

Given the potential for adverse health outcomes, many studies have demonstrated the impact of flavoring chemicals on carbonyl production during vaping [11,12,32]. Khlystov and Samburova [11] compared aldehydes in vaping emissions of flavored and unflavored e-liquids and reported that flavored e-liquids dominated the production of aldehydes (e.g., formaldehyde, acetaldehyde, acrolein, glyoxal, benzaldehyde, propionaldehyde, and m-tolualdehyde). Gillman et al. [12] found aldehyde emission levels were 150–200% higher in flavored e-liquids than unflavored e-liquids. Nevertheless, the formation of carbonyls that are not typically monitored via other reaction routes, such as direct oxidation or degradation of alcohol-containing flavoring compounds, has received less attention.

In this study, we aimed to characterize reactive carbonyl species emitted via thermal decomposition and direct oxidation of alcohol moieties in e-liquid ingredients during vaping. The e-liquid base (PG and VG) and four flavored e-liquids (*trans*-2-hexenol, benzyl alcohol, l-(-)-menthol, and linalool) were prepared in house. We hypothesized that during vaping, *trans*-2-hexenol, benzyl alcohol, l-(-)-menthol and linalool could produce unique carbonyls, such as *trans*-2-hexenal, benzaldehyde, menthone, and linalool-8-aldehyde (6-hydroxy-2,6-dimethylocta-2,7-dienal), respectively. Emitted carbonyls from flavored and unflavored e-liquids were identified by O-(2,3,4,5,6-pentafluorobenzyl)hydroxylamine (PFBHA) derivatization coupled to solid-phase microextraction (SPME) using gas chromatography/electron ionization-mass spectrometry (GC/EI-MS). Carbonyl levels were quantified using the 2,4-dinitrophenylhydrazine (2,4-DNPH) derivatization method followed by liquid chromatography/electrospray ionization-quadrupole time-of-flight mass spectrometry (LC/EIS-QTOFMS). To investigate the health implications of carbonyls associated with vaping, formation of carbonyl-GSH adducts were measured using LC/ESI-QTOFMS. The global and local reactivity parameters of selected carbonyl compounds were calculated using computational chemistry methods to predict their chemical reactivity and potential toxicity.

## 2. Materials and Methods

### 2.1. Chemicals

Most e-liquids contain flavoring chemicals, and alcohol-containing chemicals are a class of compounds that are frequently found in e-liquids [33,34,35]. Four alcohol-containing flavoring chemicals selected for this study were *trans*-2-hexenol (97%, Alfa Aesar, Ward Hill, MA, USA), benzyl alcohol (>95%, Fisher Science Education, Hanover Park, IL, USA), l-(-)-menthol (99.5%, Acros Organics, Fair Lawn, NJ, USA), and linalool (>96%, TCI America, Portland, OR, USA). Omaiye et al. [33] reported the occurrence of menthol in 50% of 227 e-cigarette refill fluid samples and found that linalool and benzyl alcohol ranked 4th and 11th as frequently occurring flavoring chemicals in sampled refill fluids. In addition, 2-hexenol has been identified as an alcohol flavoring chemical used in e-liquids [35]. Different flavoring chemicals provide a variety of aromas and tastes: menthol is known to have a minty aroma [36]; benzyl alcohol has a sweet or almond-fruity taste [34]; linalool creates a sweet floral-woody and slight citrus flavor [34]; and 2-hexenol gives an apple or fruity flavor [37].

PG (>99%, TCI America) and VG (99.9%, Fisher Chemical) were used as e-liquid base solvents to prepare the unflavored e-liquid formulation. PFBHA (98%, Acros Organics), 2,4-DNPH (97%, Sigma-Aldrich, St. Louis, MO, USA), carbonyl-2,4-DNPH standard mixture (certified reference material in acetonitrile, Sigma-Aldrich), HPLC grade acetonitrile (≥99.9%, Fisher Chemical), methanol (HPLC Grade, 99.9%, Fisher Chemical), and 6N hydrochloric acid (HCl, Fisher Chemical) were purchased for identification and quantification of carbonyls in vaping emissions. L-glutathione (GSH, 97%, Alfa Aesar) were purchased to analyze potential adduct formations between GSH and selected carbonyls: formaldehyde (37%, Fisher Chemical) and *trans*-2-hexenal (97%, TCI America).

### 2.2. E-Cigarette Device, Cartridges, and E-Liquid Preparation

A Vapros Spinner II (124.5 mm × 16.5 mm) vape pen was used for the vaping experiments in this study. The pen consists of 1600 mAh battery capacity, a charging voltage of 4.2 V/420 mAh, and a variable voltage dial (3.3–4.8 V). The vape pen was operated at 3.8 V for all vaping experiments. CCell® cartridges (TH2 510 thread, Shenzhen, China) with 0.5-mL volume capacity and resistance of 2.1 ohm were used.

K-type thermocouple wires (MN Measurement Instruments, Saint Paul, MN, USA) and data logger (Mo. SDL200, Extech, Nashua, NH, USA) were used to measure and record temperatures of the coil and PG-VG e-liquid in the cartridge (mouthpiece uncapped) during a vaping experiment to assess temperatures of the coil and e-liquids during the actual vaping and emission collection experiments. The e-cigarette pen was preconditioned with 25 puffs (as 1 cycle) before temperature measurements. Appendix A shows recorded temperature profiles of the coil and PG-VG e-liquid during vaping, with a 4-s puff and 25-s inter-puff interval. Between each cycle, the e-cigarette pen was rested for 5 min. Four cycles of puffing measurements were recorded. The highest temperatures of the coil and PG-VG during vaping were 149 ± 32 °C and 43 ± 6 °C, respectively. During inter-puff intervals, the baseline temperatures reached by the coil, and PG-VG were 26 °C and 25 °C, respectively. The room temperature remained constant at 20–21 °C.

The e-liquid formulations were prepared by mixing PG-VG base solvents and the flavoring chemicals in-house. First, all e-liquids’ base solvents consisted of a mixture of PG (70%) and VG (30%) by volume. Each flavored e-liquid contained 90% of PG-VG and 10% of an alcohol-containing flavoring compound (*v*/*v*). The unflavored e-liquid contained only the PG-VG mixture.

### 2.3. Identification of Carbonyls Emitted from Vaping

One objective of this study was to identify carbonyls in vaping emissions produced from thermal degradation and direct oxidation of alcohol-containing flavoring chemicals during vaping. To collect the vaping emissions, the e-cigarette pen was attached to a cold trap apparatus positioned inside a box filled with dry ice (Figure 1). The output of the cold trap was connected to a diaphragm pump (Gast Manufacturing Inc., Benton Harbor, MI, USA). A critical orifice was used to control the flow at ~0.46 L per minute. Before the actual collection of vaping emissions, the e-cigarette pen was preconditioned with at least 25 puffs to avoid artifacts from dry puffs. Then, 4 cycles of 25 puffs (i.e., total 100 puffs) of vaping emissions were collected at a 4-s puff and 25-s inter-puff interval using a programmable timer switch (Nearpow, model T319). This vaping topography is within the recommended range by Farsalinos et al. [38]. For every 25 puffs (as a cycle), the vape pen was rested for 5 min.

The cold trap sample was then treated with 1.5 mL of 1 mM PFBHA solution (dissolved in MiliQ water). The sample was stored at room temperature in the dark for 24 h to allow PFBHA to react with carbonyl compounds in the vaping emission samples. PFBHA can react with carbonyl compounds to form oxime derivatives, which can be analyzed using GC-MS methods [39]. After 24 h, SPME fibers (65 μm PDMS/DVB fused silica, Supelco) were used to extract the PFBHA-carbonyl oxime products from the sample solutions. The SPME fiber was submerged into each sample for 60 min and used for GC/EI-MS analysis immediately.

The collected carbonyl-PFBHA analytes on SPME fibers were analyzed using a GC/EI-MS system (Agilent 6890N GC and 5975C MSD). The SPME fibers were manually injected into the GC inlet in splitless mode and desorbed for at least 2 min at 250 °C. Analytes desorbed from the SPME fibers were separated by an Agilent Technologies DB-5MS column (30 m × 0.25 mm i.d., 0.25-μm film). The flow rate of the carrier gas (i.e., helium) was 1 mL/min. The GC temperature gradient was set as follows: an initial temperature at 60 °C held for 4 min, followed by a temperature increase to 90 °C at a rate of 15 °C min^−1^ and held for 4 min, followed by a temperature increase to 250 °C at a rate of 10 °C min^−1^, and held for 5 min. The total run time was 31 min. Data acquisition was performed in full-scan mode within *m*/*z* 30–500.

### 2.4. Quantification of Carbonyls in Vaping Emissions of E-Liquids

Derivatization of vaping emissions with 2,4-DNPH was used to estimate the levels of seven target carbonyls (formaldehyde, acetaldehyde, acrolein, acetone, propionaldehyde, butyraldehyde, and benzaldehyde). The 2,4-DNPH derivatization method used was modified from a previously published paper [40]. Vaping emissions (total 25 puffs) from each e-liquid were collected using the same cold trap method and vaping topography described in Section 2.3. The vaping samples were reacted with 1 mL of 6.5 mM 2,4-DNPH solution (acidified with HCl in acetonitrile) for 90 min at 50 °C. Afterwards, each vaping emission sample was aliquoted into three amber vials and stored at 4 °C before LC/ESI-QTOFMS analysis. A carbonyl-2,4-DNPH mixture standard (certified reference material in acetonitrile) containing the target carbonyls was used to quantify the levels of interested carbonyls in emission samples.

An Agilent Technologies 6545 LC/Q-TOFMS system equipped with an ESI source was utilized for analysis. The diluted carbonyl-2,4-DNPH standard mixture contained 150 ng/µL formaldehyde-2,4-DNPH, 100 ng/µL acetaldehyde-2,4-DNPH, and 50 ng/µL acrolein-, acetone-, propionaldehyde-, butyraldehyde-, and benzaldehyde-2,4-DNPH. The sample mixtures were separated by a Poroshell 120 EC-C18 column (3 × 100 mm, 2.7 µm). The mobile phase A was 0.1% formic acid in water, and mobile phase B was 0.1% formic acid in acetonitrile. The mobile phase gradient was 35% B–35% B from 0 min to 5 min, 35% B–70% B from 5 min to 12 min, and 70% B–70% B from 12 min to 14 min. The post-run time was 5 min. The sample injection volume was 20 µL, and the instrument was operated in the negative ion mode with a 0.5 mL min^−1^ flow rate.

### 2.5. Carbonyl-GSH Adduct Formation

Carbonyls are reactive compounds that can form adducts with nucleophiles such as GSH. The carbonyl-GSH adducts were examined to evaluate the health implications of vaping-associated carbonyls. First, formaldehyde and *trans*-2-hexenal sample solutions were prepared with 5 μL of their standards in 1 mL methanol. Then, each reaction mixture contained 50 μL of the carbonyl sample solution, 100 μL of GSH (28.4 mM in MiliQ water), and 1200 μL MiliQ water. The GSH blank sample contained only 100 μL of GSH and 1250 μL MiliQ water. Each reaction mixture was incubated at 37 °C for 120 min, then stored in a −20 °C freezer until LC/EIS-QTOFMS analysis.

The instrument used to analyze the carbonyl-GSH samples was an Agilent Technologies 6545 LC/Q-TOFMS system equipped with an ESI source. The sample mixtures were separated by a Poroshell 120 EC-C18 column (3 × 100 mm, 2.7 µm). The mobile phase A was 0.1% formic acid in water, and mobile phase B was 0.1% formic acid in acetonitrile. The mobile phase gradient was 0% B–100% B from 2 min to 18 min. The column was washed with 100% B and equilibrated with 100% A before running the next gradient. The sample injection volume was 30 µL, and the instrument was operated in the positive ion mode.

### 2.6. Global Electrophilicity and Local Site Reactivity Calculations

The global electrophilicity index, *ω*, a reactivity parameter introduced by Parr et al. in 1999 describes the stabilization energy of atoms or molecules when they accept an additional electronic charge from the environment [26]. In this study, Gaussian 16W was used to optimize geometries and compute *ω* of interested simple carbonyls (i.e., formaldehyde, acetaldehyde, and benzaldehyde) and α,β-unsaturated carbonyls (i.e., acrolein and *trans*-2-hexenal). Density functional theory was used to optimize the geometry of molecules at the B3LYP/6-311 + G(d,p) level of theory, and the conductor-like polarizable continuum model (CPCM) was implemented to include water as a solvent. The optimized geometries are shown in Appendix A.

To calculate *ω*, chemical potential (*μ*) and chemical hardness (*ƞ*), which are associated with ionization potential (*I*) and electron affinity (*A*) [26,41], were calculated using the finite difference approximation. Ionization potential and electron affinity energies were obtained using energies of compounds in their neutral (charge 0), cationic (charge +1), and anionic (charge −1) states [42], which were computed using natural population analysis (NPA) in Gaussian 16W. Ionization potential was calculated as E_cation_ − E_neutral_ (Equation (1)), while electron affinity was calculated as E_neutral_ − E_anion_ (Equation (2)). Calculations for chemical potential, chemical hardness, and global electrophilicity index are shown in the following equations [26,43].
(1)$$I=E_{cation}-E_{neutral}$$
(2)$$A=E_{neutral}-E_{anion}$$
(3)$$\mu =-\frac{I+A}{2}$$
(4)$$\eta =\frac{I-A}{2}$$
(5)$$\omega =\frac{\mu^{2}}{2\eta }$$

In addition to global electrophilicity calculations, the condensed Fukui functions that represent local reactivity at specific sites within a molecule were also calculated. First, Gaussian 16W was used to optimize geometries at the B3LYP/6-311 + G(d,p) level of theory in water using the CPCM solvation model. Then, the NPA in charges 0, −1, and +1 were calculated in Gaussian 16W. Afterwards, the output values (Appendix A) were uploaded into the UCA-FUKUI 1.0 software (https://uca-fukui.software.informer.com/1.0/ (accessed on 11 March 2019)) to calculate the condensed Fukui functions ($f_{k}^{0}$, $f_{k}^{-}$, $f_{k}^{+}$, and dual-descriptor) using the finite difference method [44]. The following equations show the condensed Fukui functions for the *k*th atom in the molecule.
(6)$$f_{k}^{-}=q_{k}\left(N_{0}\right)-q_{k}(N_{0}-1),\;\mathrm{for}\;\mathrm{electrophilic}\;\mathrm{attack}$$
(7)$$f_{k}^{+}=q_{k}(N_{0}+1)-q_{k}\left(N_{0}\right),\;\mathrm{for}\;\mathrm{nucleophilic}\;\mathrm{attack}$$
(8)$$f_{k}^{0}=\frac{1}{2}\;\left(f_{k}^{+}+f_{k}^{-}\right),\;\mathrm{for}\;\mathrm{neutral}\;(\mathrm{or}\;\mathrm{radical})\;\mathrm{attack}$$
(9)$$\mathrm{Dual-}\mathrm{descriptor}=f_{k}^{+}-f_{k}^{-}$$

In the above equations, $q_{k}(N_{0}-1)$, $q_{k}(N_{0}+1)$, and $q_{k}\left(N_{0}\right)$ represent the number of electrons associated with the *k*th atom in the molecule, where the total number of the electrons in the molecule is *N*_0_ − 1, *N*_0_ + 1, and *N*_0_ [44]. The dual-descriptor is the difference between $f_{k}^{+}$ and $f_{k}^{-}$, and it is either >0 or <0 [44,45,46]. If the dual-descriptor is >0, it indicates that the reaction site favors a nucleophilic attack, and if the dual-descriptor is <0, an electrophilic attack may be favored [44].

## 3. Results and Discussions

### 3.1. Identification of Carbonyls in Vaping Emissions

PFBHA derivatization was used to identify carbonyls emitted from vaping of flavored and unflavored e-liquids, followed by subsequent SPME fiber extraction and GC/EI-MS analysis. Some carbonyl-PFBHA oxime derivatives have two possible stereoisomers (E and Z) [39]. Figure 1 compares extracted ion chromatograms (EICs) of *m*/*z* 181, an abundant fragment ion of carbonyl-PFBHA oxime derivatives, from vaping emission samples of unflavored (i.e., PG-VG only) and flavored e-liquids. EICs reveal that while all emissions samples share some common peaks, there are also distinct peaks among samples.

Using the NIST mass spectral library search (NIST MS Search 2.0), Table 1 displays 14 carbonyls identified using their signature ions (e.g., *m*/*z* 181 and the molecular ion of carbonyl-PFBHA oxime derivatives). The derivatization reaction between each identified carbonyl and PFBHA, and their oxime derivatives are shown in Appendix A. The identified carbonyls are formaldehyde, acetaldehyde, acetone, propionaldehyde, acrolein, butyraldehyde, glyoxal, methylglyoxal, dimethylglyoxal, *trans*-2-hexenal, benzaldehyde, *trans*-2-methyl-butenal, 3-methyl-2-butenal, and 2-methyl-2-pentenal. At least one of the identified carbonyls was present in vaping emissions of one e-liquid.

Table 1 shows that low molecular weight carbonyls, such as formaldehyde, acetaldehyde, acetone, glyoxal, and methylglyoxal, were detected in vaping emissions of PG-VG and four flavored e-liquids. Propionaldehyde and acrolein were detected in emission samples of PG-VG, *trans*-2-hexenol-flavored, and linalool-flavored e-liquids. Benzaldehyde and 3-methyl-2-butenal were present in vaping emissions of benzyl alcohol- and linalool-flavored e-liquids, respectively. *trans*-2-Hexenal was detected in emissions of e-liquids flavored with *trans*-2-hexenol, l-(-)-menthol, or linalool. Additionally, butyraldehyde and 2-methyl-2-pentenal were detected in emissions of the *trans*-2-hexenol-flavored e-liquid. Propionaldehyde, acrolein, and/or dimethylglyoxal, on the other hand, were not detected in the vaping emissions of benzyl alcohol- and l-(-)-menthol-flavored e-liquids. One possible explanation is that benzyl alcohol and l-(-)-menthol have antioxidative properties [47,48]. For benzyl alcohol, the oxidation reaction was most likely directed toward the formation of benzaldehyde.

Several carbonyls (formaldehyde, acetaldehyde, acetone, propionaldehyde, acrolein, glyoxal, methylglyoxal, and benzaldehyde) identified in this study are consistent to those reported previously [10,49,50,51,52]. While formaldehyde, acetaldehyde, and acetone are common products of PG and VG degradation processes, productions of benzaldehyde and *trans*-2-hexenal are very likely linked to transformation of flavoring chemicals in e-liquids. For instance, Kosmider et al. [52] detected benzaldehyde in 108 out of 145 flavored e-liquids’ vaping aerosols, and the highest benzaldehyde level (141.2 µg/30 puffs) was found in emissions of cherry-flavored e-liquids [52].

Interestingly, *trans*-2-hexenal was detected in vaping emissions of *trans*-2-hexenol (a C6 alcohol), l-(-)-menthol (a C10 monoterpene), and linalool (a C10 monoterpenoid). It is well known that primary alcohols can oxidize to yield aldehydes. Thus, it is reasonable to detect *trans*-2-hexenal in *trans*-2-hexenol emissions. The formation of *trans*-2-hexenal in menthol- and linalool-flavored e-liquid vaping emissions could be attributed to oxidation and thermal degradation processes, as described in previous literature [53,54,55]. However, future studies are needed to investigate the detailed pathway of formation of *trans*-2-hexenal. Additionally, the presence of methylbutenal and methylpentenal compounds in the emissions of some flavored e-liquids may indicate that they were formed through oxidation and degradation of parent flavoring chemicals.

E-cigarette emission products are influenced by different parameters, including e-cigarette devices, battery voltage, vaping topography, and e-liquid constituents. Sleiman et al. [51] demonstrated acetol and 2-propenol were primarily produced by PG degradation, and acrolein was mostly produced by VG degradation. Kosmider et al. [56] reported 4–200 times higher formaldehyde, acetaldehyde, and acetone levels as battery voltage increased from 3.2 to 4.8 V. A combination of factors such as lower battery voltage output, PG/VG ratio, and e-liquid constituents may explain why hypothesized production of menthone and linalool-8-aldehyde were not detected in the vaping emissions of e-liquids flavored with l-(-)-menthol and linalool, respectively.

### 3.2. Levels of Target Carbonyls in E-Liquids’ Vaping Emissions

The 2,4-DNPH method was used to estimate the amount of seven target carbonyls in vaping emissions of PG-VG and four flavored e-liquids. Carbonyls can react with 2,4-DNPH to form hydrazone derivatives [57,58], which can be identified by their EICs *m*/*z* of [M-H]^−^ from LC/ESI-Q-TOFMS analysis. The *m*/*z* values of hydrazone derivatives of target carbonyls are 209 (formaldehyde), 223 (acetaldehyde), 235 (acrolein), 237 (acetone and propionaldehyde), 251 (butyraldehyde), and 285 (benzaldehyde).

As shown in Figure 2, formaldehyde, acetaldehyde, and acetone/propionaldehyde were detected in emissions samples from PG-VG and flavored e-liquids. In the case of formaldehyde, the amounts per puff were as follows (ranking from the highest to the lowest): 22,717 ± 4036 ng/puff (PG-VG), 1072 ± 319 ng/puff (*trans*-2-hexenol), 1248 ± 142 ng/puff (l-(-)-menthol), 399 ± 64 ng/puff (benzyl alcohol), and 126 ± 44 ng/puff (linalool). For acetaldehyde, PG-VG and *trans*-2-hexenol had two highest amounts of 4082 ± 42 ng/puff and 85 ± 3 ng/puff, respectively. The rest of the e-liquid emissions had <20 ng/puff of acetaldehyde detected. As for acetone and propionaldehyde, their amounts in vaping emission samples were combined, as they have the same EIC of *m*/*z* 237 and close retention times that could not be separated by our LC method. The order of acetone/propionaldehyde ranking from the highest to the lowest amount was 401 ± 152 ng/puff (l-(-)-menthol), 382 ± 104 ng/puff (*trans*-2-hexenol), 360 ± 95 ng/puff (benzyl alcohol), 314 ± 138 ng/puff (linalool), and 216 ± 33 ng/puff (PG-VG).

Carbonyls not detected in all e-liquids’ emission samples were acrolein, butyraldehyde, and benzaldehyde. Acrolein was detected in emission samples of PG-VG (12 ± 2 ng/puff) and *trans*-2-hexenol (2 ± 1 ng/puff). Butyraldehyde was detected in emissions of *trans*-2-hexenol (35 ± 6 ng/puff), benzyl alcohol (1 ± 0.1 ng/puff), and l-(-)-menthol (0.4 ± 0.1 ng/puff). On the other hand, benzaldehyde (0.7 ± 0.1 ng/puff) was only detected in emission samples of benzyl alcohol. The levels of formaldehyde, acetaldehyde, and acrolein were lower in vaping emissions of alcohol-flavored e-liquids. The addition of alcohol flavoring chemicals in e-liquids (with PG-VG as the base solvent) could potentially lead to competing reactions during thermal decomposition [59,60]. For example, intermediates produced from the vaping of flavored e-liquids might interact with each other and shift the product distribution when compared to the vaping of PG-VG e-liquid alone. However, the detailed mechanism needs further investigation.

E-liquid constituents, among many parameters, affects carbonyl concentrations and chemical composition in vaping emissions. Figure 2 shows presence of formaldehyde, acetaldehyde, acrolein, acetone, propionaldehyde, and butyraldehyde in vaping emission samples, which are consistent with published findings [10,11,12,13]. In addition, composition of emitted carbonyls may be largely contributed to their parent compounds. Sleiman et al. [51] examined acrolein that was mostly produced from VG degradation, while formaldehyde was produced from both PG and VG degradation. Our results show higher concentrations of formaldehyde emitted compared to acrolein. In addition, presence of benzaldehyde in benzyl alcohol e-liquid’s emission samples may explain primary production of benzaldehyde from benzyl alcohol degradation.

Addition of flavoring chemicals in e-liquids can also affect carbonyl production. Study conducted by Khlystov and Samburova [11] shows higher concentrations of carbonyls in vaping aerosols of flavored e-liquids than unflavored e-liquids. In this study, unflavored e-liquids (PG-VG) produced higher levels of formaldehyde, acetaldehyde, and acrolein than flavored e-liquids. Yet, all flavored e-liquids produced higher levels of acetone/propionaldehyde. Moreover, at least one flavored e-liquid produced butyraldehyde or benzaldehyde; these compounds were undetected or below detection limit (S/N < 3) in PG-VG emission samples. For example, *trans*-2-hexenol-flavored e-liquid emitted the highest levels of butyraldehyde. Oxidative decomposition could be a possible pathway contributing to the increased formation of butyraldehyde (C_4_H_8_O) from the degradation of *trans*-2-hexenol (C_6_H_12_O) in the presence of oxygen [53,54,55]. This shows degradation of flavoring chemicals in e-liquids during vaping leads to additional production of carbonyls.

### 3.3. Carbonyl-GSH Adduct Detection

Carbonyls are electrophiles that can form adducts with thiol-containing compounds via 1,2-carbonyl addition (Schiff base formation) or 1,4-conjugated addition (or Michael-type addition) [28]. A target search for potential formation of adducts between two selected carbonyls (formaldehyde and *trans*-2-hexenal) and GSH was conducted as shown in Figure 3. Formaldehyde as a simple carbonyl (with only one active site for nucleophilic attack, i.e., C_carbonyl_) can undergo 1,2-addition to form adducts with GSH. To compare, *trans*-2-hexenal as an α,β-unsaturated carbonyl can form adducts with GSH via both 1,2-carbonyl (nucleophilic attack on C_carbonyl_) and 1,4-conjugated (nucleophilic attack on C_β_) addition reactions. Analytes represented as [M+H]^+^ were used to plot EICs of formaldehyde-GSH (*m*/*z* 338) and *trans*-2-hexenal-GSH (*m*/*z* 408).

### 3.4. Global Electrophilicity and Local Site Reactivities of Carbonyls

Global electrophilicity and local site reactivities of formaldehyde, acetaldehyde, benzaldehyde, acrolein, and *trans*-2-hexenal were calculated using computational chemistry approaches. As shown in Table 2, the global reactivity descriptors (*I*, *A*, *μ*, *η*, and *ω*) were calculated using the density functional theory. Values of *I* and *A* were used to calculate *μ, η,* and *ω* (Equations (3)–(5)), which are important for assessing global reactivity of carbonyls. Typically, the higher the *ω* value, the more electrophilic the compound. When comparing the *ω* values for simple carbonyls, benzaldehyde has the highest *ω* value (10.43 eV), followed by formaldehyde (8.62 eV) and acetaldehyde (7.46 eV). For α,β-unsaturated carbonyls, acrolein has higher *ω* (10.57 eV) than *trans*-2-hexenal (8.41 eV). As *ω* works well to predict the overall reactivity of carbonyl compounds, condensed Fukui parameters are preferred to model compounds with multiple reactive sites (e.g., α,β-unsaturated carbonyls).

Condensed Fukui parameters ($f_{k}^{0}$, $f_{k}^{-}$, $f_{k}^{+}$, and dual-descriptor) are useful for predicting local site reactivities of α,β-unsaturated carbonyls. $f_{k}^{0}$ is used to represent a neutral or radical attack, $f_{k}^{-}$ represents an electrophilic attack, $f_{k}^{+}$ represents a nucleophilic attack, and the dual-descriptor represents an electrophilic (value < 0) or nucleophilic (value > 0) attack [44]. Table 3 shows the calculated $f_{k}^{+}$ values for α-carbon (C_α_), β-carbon (C_β_), carbonyl-carbon (C_carbonyl_), and carbonyl-oxygen (O_carbonyl_) for different carbonyls listed. The calculated $f_{k}^{0}$, $f_{k}^{-}$ and the dual-descriptor values are shown in Appendix A. The condensed Fukui functions for C*_α_* and C*_β_* are only applicable to α,β-unsaturated carbonyls.

For simple carbonyls, C_carbonyl_
$f_{k}^{+}$ values are greater than O_carbonyl_ values (Table 3). This represents that the simple carbonyls can form adducts with nucleophiles (e.g., GSH) via 1,2-carbonyl addition. For most of the α,β-unsaturated carbonyls, C_β_ has highest $f_{k}^{+}$ values, followed by C_carbonyl_, O_carbonyl_, and C_α_. Acrolein’s O_carbonyl_ has slightly higher $f_{k}^{+}$ value than C_carbonyl_. These results support that α,β-unsaturated carbonyls can likely form adducts with GSH (or thiol containing biomolecules) through both 1,4-conjugated and 1,2-carbony additions.

When examining the dual-descriptor values (difference between $f_{k}^{+}$ and $f_{k}^{-}$) of simple carbonyls in Appendix A, all C_carbonyl_ values are <0 (electrophilic attack) and O_carbonyl_ values are >0 (nucleophilic attack). For all α,β-unsaturated carbonyls, their C_carbonyl_ and C_α_ dual-descriptor values are >0. Both acrolein and *trans*-2-hexenal have O_carbonyl_ < 0 and C_β_ > 0.

### 3.5. Potential Limitations

Some potential limitations in this study should be noted. First, emissions of carbonyls during vaping are influenced by multiple factors, including types of e-cigarette device used, battery voltage, PG/VG ratios, levels of flavoring chemicals in e-liquids, and vaping topography [51,61,62]. These factors make it difficult to directly compare our results of carbonyl emissions with previously published work. For instance, studies have shown that flavored e-liquids are linked to increased levels of carbonyl emissions, such as formaldehyde, acetaldehyde, and propionaldehyde [11,12]. Results from this study (Figure 2) do not show increased levels of formaldehyde and acetaldehyde in vaping emissions of flavored e-liquids, except for acetone/propionaldehyde. Furthermore, our house-made e-liquids do not reassemble commercially available e-cigarette products, which frequently contain a complex mixture of base solvents (e.g., PG/VG at different ratios), nicotine, flavoring chemicals, and other ingredients [10,33,63]. Our e-liquid formulations contained only PG-VG and flavoring chemicals for direct comparison of carbonyl emissions between PG-VG and flavored e-liquids. Additional research is needed to better understand emissions of carbonyls and other toxic compounds during vaping using commercially available e-liquids. Finally, the carbonyls listed in Table 1 analyzed by GC/EI-MS were identified using their signature ions. To confirm the presence of these compounds in emission samples, authentic standards are required.

## 4. Conclusions and Implications

Using GC/EI-MS and LC/ESI-QTOFMS methods, this study characterized 14 carbonyls and levels of seven target carbonyls in vaping emissions of flavored and unflavored e-liquids. The identified carbonyls are formaldehyde, acetaldehyde, acetone, propionaldehyde, acrolein, butyraldehyde, glyoxal, methylglyoxal, dimethylglyoxal, n-butanal, 3-methyl-2-butenal, *trans*-2-methyl-2-butenal, 2-methyl-2-pentenal, *trans*-2-hexenal, and benzaldehyde.

This study demonstrated the formation of commonly reported toxic carbonyls (e.g., formaldehyde, acetaldehyde, acetone, propionaldehyde, and glyoxal) and unique carbonyls associated with transformation of flavoring chemicals during vaping (e.g., *trans*-2-hexenal and benzaldehyde). Formaldehyde-GSH and *trans*-2-hexenal-GSH adducts formed via 1,2-additon and/or 1,4-conjugated addition reactions between carbonyls and GSH highlighted vaping-associated reactive carbonyl species as an external source of toxicity. The global electrophilicity indices (ω) show benzaldehyde, acrolein, and *trans*-2-hexenal as most reactive among selected carbonyls, while the local reactivity parameter, $f_{k}^{+}$, supports that C_carbonyl_ of simple carbonyls, and C_carbonyl_ and C_β_ of α,β-unsaturated carbonyls are more favorable for nucleophilic attacks. A wide variety of carbonyl compounds have been detected in vaping emissions, but only a few have been reported. These emitted carbonyls may pose potential health risks upon inhalation depending on their abundance in the vaping emissions as well as their chemical reactivity. Further studies are required to comprehensively profile the emission products from flavoring chemicals to gain a better understanding of potential health effects associated with vaping.

## Data Availability

The data presented in this study are available in this article and its Appendix A file.

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
