# Peer review of "Carbonyl Composition and Electrophilicity in Vaping Emissions of Flavored and Unflavored E-Liquids"

_toxics, 2021, doi:10.3390/toxics9120345_

Round 1
Reviewer 1 Report
The review has been included in a file.

Reviewer 2 Report
In this manuscript, the authors have investigated the effect of vaping e-liquid constitution on the emitted carbonyls. They have used GC-MS and LC-MS for the characterization and determination of these carbonyls. The results found are interesting; however, the results need more discussion and clarifications. Hence, there are some points needed to be addressed prior to the consideration of the publication of this manuscript.
- Why have the authors omitted nicotine from vaping e-liquid in their study?
- In section 2.3., it is difficult to imagine the trapping process from the text. Please draw a diagram for the trapping device.
- In table 1, why in case of benzyl alcohol flavored e-liquid propionaldehyde, acrolein, and dimethyl glyoxal were not detected. Does benzyl alcohol act as an antioxidant, and the oxidation reaction was directed only to the formation of benzaldehyde? Also, why in the case of menthol-flavored e-liquid propionaldehyde and acrolein are not detected.
- In table 1 also, what is the possible pathway of formation of trans-2-hexenal in menthol and linalool flavored e-liquids.
- In Fig. 2, why the presence of alcohol flavoring liquids decreases the formation of formaldehyde, acetaldehyde, and acrolein?
- In Fig. 2 also, why the presence of trans-2-hexenol largely increased the formation of butyraldehyde. What is the possible oxidation pathway for this butyraldehyde formation from trans-2-hexenol?
- In your study, menthone and linalool-8-aldehyde were not detected; thus, there is no any importance for the computational study of their reactivity. Hence they should be removed from the computational study in section 3.4.
- Do not use abbreviations (such as PG-VG) in keywords.
Round 2
Reviewer 2 Report
The authors have adequately addressed all my comments and the discussion of the results on the manuscript becomes acceptable. The paper could be accepted in its current form.